# Tackling Food Insecurity in Cabo Verde Islands: The Nutritional, Agricultural and Environmental Values of the Legume Species

**DOI:** 10.3390/foods10020206

**Published:** 2021-01-20

**Authors:** Miguel Brilhante, Eromise Varela, Anyse P. Essoh, Arlindo Fortes, Maria Cristina Duarte, Filipa Monteiro, Vladimir Ferreira, Augusto Manuel Correia, Maria Paula Duarte, Maria M. Romeiras

**Affiliations:** 1Linking Landscape, Environment, Agriculture and Food (LEAF), Instituto Superior de Agronomia (ISA), Universidade de Lisboa, Tapada da Ajuda, 1340-017 Lisboa, Portugal; miguelbrilhante131@hotmail.com (M.B.); varelaeromise@gmail.com (E.V.); anyse.sofia@gmail.com (A.P.E.); fimonteiro@fc.ul.pt (F.M.); 2Centre for Ecology, Evolution and Environmental Changes (cE3c), Faculdade de Ciências, Universidade de Lisboa, Campo Grande, 1749-016 Lisboa, Portugal; mcduarte@fc.ul.pt; 3Research Centre in Biodiversity and Genetic Resources (CIBIO), InBIO Associate Laboratory, Pole of Azores, Faculdade de Ciências e Tecnologia, Universidade dos Açores, 9500-321 Ponta Delgada, Portugal; 4Nova School of Business and Economics, Campus de Carcavelos, 2775-405 Carcavelos, Portugal; 5Escola Superior de Ciências Agrárias e Ambientais, Universidade de Cabo Verde, Santiago, Praia CP 379, Cape Verde; arlindo.fortes@docente.unicv.edu.cv (A.F.); vladmir.ferreira@adm.unicv.edu.cv (V.F.); 6Centro de Estudos sobre África para o Desenvolvimento (CEsA), Instituto Superior de Economia e Gestão, Universidade de Lisboa, 1200-781 Lisboa, Portugal; 7Centre of Tropical Studies for Development (CENTROP), Instituto Superior de Agronomia (ISA), Universidade de Lisboa, 1349-017 Lisboa, Portugal; amcorreia@isa.ulisboa.pt; 8MEtRICs/DCTB, Faculdade de Ciências e Tecnologia, Universidade NOVA de Lisboa, 2829-516 Caparica, Portugal

**Keywords:** tropical dry islands, middle income countries (MICs), legumes diversity, phenolic contents, mineral content, nutritional composition, agronomic value

## Abstract

Legume species are important food sources to reduce hunger and deal with malnutrition; they also play a crucial role in sustainable agriculture in the tropical dry islands of Cabo Verde. To improve the knowledge of the heritage of plant genetic resources in this Middle Income Country, this study had three main goals: (i) to provide a checklist of food legumes; (ii) to investigate which species are traded in local markets and, based on field surveys, to compare species for their chemical, phenolic, antioxidant, and nutritional composition; and (iii) to discuss the agronomic value and contribution to food security in this archipelago. Our results revealed that 15 species are used as food and 5 of them are locally traded (*Cajanus cajan*, *Lablab purpureus*, *Phaseolus lunatus*, *Phaseolus vulgaris*, and *Vigna unguiculata*). The role of these species as sources of important minerals, antioxidants, and nutritional components for food security is highlighted, and the native ones (*Lablab purpureus* and *Vigna unguiculata*) stand-out as particularly well-adapted to the climate of these islands, which are already experiencing the adverse effects of climate change. We conclude that the sustainable use of these genetic resources can contribute to the reduction of hunger and poverty, thus meeting some challenges of the Sustainable Development Goals.

## 1. Introduction

Agricultural development is imperative to improve food security and nutrition [1]. Increasing the quantity and diversity of food will provide the primary source of income for many people, which is particularly important in low- and middle-income countries (LMICs), namely those in the African continent. The Food and Agriculture Organization (FAO) recently assessed the list of countries undergoing food emergency and in need of external assistance for food, where 34 out of 45 countries are in Africa [2]. The effects of the COVID-19 pandemic, particularly through the loss of income and jobs related to confinement measures, severely aggravated global food security conditions (e.g., [3]). These are even more worrying in countries where food security was already a major concern, such as the tropical dry islands of Cabo Verde where the agriculture sector is extremely limited by natural constraints such as drought periods, poor soils, scarcity of cropland and low technological level of implementation [4].

The Cabo Verde archipelago comprises the southernmost islands of Macaronesia (i.e., Azores, Madeira, Savage Islands, Canary Islands, and Cabo Verde) and it is located in the Sahelian arid and semiarid regions in close proximity to the western African coast [5]. Cabo Verde consists of nine inhabited islands grouped in Northern Islands (São Nicolau, São Vicente and Santo Antão), Eastern Islands (Sal, Boavista and Maio) and Southern Islands (Santiago, Fogo and Brava). The vascular flora comprises ca. 740 vascular plant taxa, of which ca. 92 are endemic [6]. In a recent conservation assessment based on the International Union for Conservation of Nature (IUCN) Red list criteria, ca. 78% of the endemic plant species were considered threatened, mostly as a consequence of the growing habitat degradation, human disturbance (e.g., intentional use for agriculture or traditional uses) and introduction of exotic species since the beginning of the islands’ colonization [7].

Cabo Verde was the first tropical archipelago colonized by Europeans, in 1462, and due to its geographic location in the mid-Atlantic Ocean, it became an important hub for trans-Atlantic trade routes [8]. Particularly since the 16th century, these islands played an important role as a centre of dissemination and acclimatization of tropical crop species of key economic importance, prior to their cultivation in other regions [9]. Also, during human settlement, the introduction of useful plants was of chief importance to change the present-day composition of the archipelago’s flora, which comprises a great number of exotic taxa [10]. Historical contingencies, namely the location of the main harbours (i.e., Mindelo, in São Vicente, and Cidade da Praia, in Santiago) and the establishment of the first settlements greatly impacted the knowledge of local flora and the botanical explorations in this archipelago [6].

Although several botanical explorations were performed in this archipelago since the late 18th century [11,12,13], only in 1908 was the first Agronomic Mission carried out, by P. de Lemos, Pereira da Cunha and A. Costa Andrade. In 1935, the French botanist Auguste Chevalier [14] published a seminal work “*Les Iles du Cap Vert. Géographie, biogéographie, agriculture. Flore de l’Archipel*” covering the vast majority of the plant collections made in these islands since the late 18th century. During the colonial period (1915–1974), Grandvaux Barbosa assembled the largest plant collection, and published the first comprehensive study of the territory’s agriculture [15]. This study stressed that the main rainfed crops of Cabo Verde were maize (*Zea mays* L.) and several bean species (i.e., *Cajanus cajan* (L.) Huth, *Lablab purpureus* (L.) Sweet, *Phaseolus vulgaris* L., *Phaseolus lunatus* L. and *Vigna unguiculata* (L.) Walp.). These species are still the basis of the country’s diet, the most emblematic dish of Cabo Verde being “cachupa”, which is cooked with different varieties of beans and maize. These crops are produced through rainfed subsistence farming, whereas irrigated crops, such as sugarcane, manioc and tomatoes, are mostly grown for commercial purposes. Monteiro et al. [16] reported that Santiago has the largest area used for agriculture (52.5%), followed by Santo Antão (16%) and Fogo (15.8%). These islands are better for agriculture than the others, because they have a complex variety of microclimates, ranging from more humid zones in mountain regions of Santiago (Pico da Antónia) and Santo Antão (e.g., Ribeira do Paúl), to volcanic areas in Fogo, which reaches almost 3000 m, or to lowland arid areas that experience the scourge of long-lasting droughts. During the 20th century, anthropogenic activities caused enormous damage and, particularly in humid and sub-humid areas on the North and Northeast slopes above 400 m, natural vegetation was gradually cut and destroyed and replaced by crop species [17]. Nevertheless, several conservation actions were undertaken over the last two decades by Cabo Verde authorities, in particular a system of Protected Areas (PAs) to safeguard the natural heritage of the archipelago [7,18].

Since the Convention on Biological Diversity (see: https://www.cbd.int/intro/) several steps were taken at international levels to address specific biodiversity issues, such as agrobiodiversity and sharing benefits associated with the exploitation of the plant genetic resources. Despite the key role played by the plant genetic resources in the Cabo Verdean agriculture, they are still poorly known and there is no clear understanding of which ones are relevant to the improvement of crop species, as well as their potential capacity to adapt to climatic and environmental changes. Particularly, pulses (i.e., edible seeds of plants of the legume family) have a long history as staple crops for smallholder farmers in semi-arid tropical areas of sub-Saharan Africa [19] and, as stated above, are among the most important crops of Cabo Verde [15,16].

In order to improve the knowledge of the heritage of genetic resources of the Leguminosae family in Cabo Verde, this study had three main goals. First, to provide a checklist of Leguminosae taxa used as food, and new data on their native distribution (archipelago and worldwide), common names, and other uses (i.e., forage and medicinal). Second, to investigate which legume species are consumed and traded in Santiago markets and, based on field surveys, to compare species for their chemical composition, nutritional value, and antioxidant activities. Finally, to discuss aspects concerning the agronomic value, sustainable use of pulses and other legumes, and their potential contribution to food security in this archipelago.

## 2. Materials and Methods

### 2.1. Studied Species

Data on the food Leguminosae species known in Cabo Verde were obtained through a comprehensive review of the best knowledge currently available. The baseline data for the present study was gathered from Herbarium collections [Herbarium of Instituto Superior de Agronomia of the University of Lisbon, João de Carvalho e Vasconcellos (LISI, Lisbon, Portugal); Herbarium of Instituto de Investigação Científica Tropical, University of Lisbon (LISC, Lisbon, Portugal); Herbarium of Museu Nacional de História Natural e da Ciência, University of Lisbon (LISU, Lisbon, Portugal); Herbarium of University of Coimbra (COI, Coimbra, Portugal); Herbarium of Royal Botanic Gardens, Kew (K, Richmond, UK); Herbarium of Natural History Museum (BM, London, UK); and Herbarium of Meise Botanic Garden (BR, Meise, Belgium)]. Moreover, scientific publications (e.g., [10,15,20,21,22] and online databases (e.g., Plants of the World Online [23], PROTA—Plant Resources of Tropical Africa [24], IPNI—International Plant Names Index [25] and the International Legume Database and Information Service [26]) were also accessed for information on taxonomic data, native distribution and cultivation status. We then constructed a comprehensive database including the scientific name of each species, English common names, native status in Cabo Verde, native distribution, habit and distribution in Cabo Verde.

### 2.2. Sampling

The field surveys were performed between 2018 and 2019 on Santiago Island. This island is the largest one in the archipelago, and also the one with the largest population, about a fifth of the inhabitants of Cabo Verde live on this island [27]. The climate is characteristically hot and semi-arid, with the rainy season from August to October, September being the wettest month whilst the annual average temperatures attain the maximum of 25 °C [5].

Surveys were made in the main trade markets of Santiago Island (Assomada, Calheta, Praia, São Domingos, São Jorge dos Órgãos, São Salvador do Mundo and Tarrafal) in order to identify the most cultivated and traded bean species of Cabo Verde. Data on trade species, including their origin and availability, were also obtained during the field surveys. Five species are widely consumed/traded in Santiago markets: *Cajanus cajan*, *Lablab purpureus*, *Phaseolus lunatus*, *Phaseolus vulgaris*, and *Vigna unguiculata* (Figure 1). Beans (dry seeds) from different markets were then suitably packaged and shipped by air to Portugal. Once in the laboratory (Lab. facilities of NOVA School of Science and Technology, Caparica, Portugal), these beans were subjected to chemical and nutritional analyses. A portion of each of the collected samples were germinated under controlled conditions in a greenhouse at Instituto Superior de Agronomia of the University of Lisbon (ISA/UL), in order to obtain developed and mature leaves needed to further evaluate their antioxidant capacity in the laboratory. All the species were cultivated in pots with fertile and well-drained soils with the optimal temperature for germination varying between 25 °C and 27 °C. All the seeds germinated after 10–20 days. A total of 15 accessions of beans and leaves were analysed: *Cajanus cajan* (n = 2), *Lablab purpureus* (n = 6), *Phaseolus lunatus* (n = 2), *Phaseolus vulgaris* (n = 2) and *Vigna unguiculata* (n = 3). Their origin and location in Santiago is indicated in Appendix A.

### 2.3. Chemical Composition and Antioxidant Analyses

#### 2.3.1. Reagents

Acetone, ascorbic acid, gallic acid, sodium hydroxide, potassium hydroxide, potassium sulphate, selenium metal powder, sodium carbonate and standard solutions for inductively coupled plasma optical emission spectrometry (ICP-OES) (boron, calcium, copper, iron, magnesium, manganese, phosphorus, potassium, sulphur and zinc) were from Panreac (Barcelona, Spain), 2,2-diphenyl-1-picrylhydrazyl (DPPH) was from Sigma-Aldrich (St. Louis, MO, USA) and boric acid were from Chem-lab (Zedelgem, Belgium), ethanol absolute and sulphuric acid were from Riedel-de Haën (Seelze, Germany), hydrochloric acid, nitric acid and Folin-Ciocalteu reagent were from Merck (Darmstadt, Germany). All reagents used in the analytical procedures were of analytical reagent grade. All the water used was purified using a Milli-Q water system (Millipore, Bedford, MA, USA).

#### 2.3.2. Sample Preparation

Whole beans (100 g seeds of each bean accession) were ground into flour using a stainless-steel grinder (Kunft coffee mill) and analysed for moisture, ash, protein, fibre and mineral contents.

Leaves from the different species were collected and allowed to dry for two weeks, in the dark, at room temperature. Then, dry leaves were ground into powder in a porcelain mortar with pestle. Grounded dry leaves (5 g) were extracted with 100 mL of ethanol (70% *v*/*v*), for 24 h, in the dark, at room temperature and under stirring. The resulting extracts were filtered (Whatman, n°1) and stored at—50 °C until future analyses.

#### 2.3.3. Moisture and Ash

Moisture and ash contents were determined according to the standard gravimetric method [28]. Briefly, moisture content was measured by drying the sample at 103 ± 2 °C (Laboratory heater, Memmert UL500, Schawabach, Germany) for 2 h and repeated until constant weight, while ash content was determined by incinerating the sample for 6 h at 550 ± 25 °C (Muffle furnace, Heraeus Hanau MR170E, Hanau, Germany). The analyses were performed in duplicate and the results expressed in g per 100 g wet weight.

#### 2.3.4. Crude Fibre

The crude fibre determination was performed in duplicate by Weende’s method [28]. Briefly, test portions of each sample were submitted to acid hydrolysis with 150 mL of sulfuric acid (0.128 M), at boiling temperature and for 30 min. Then, the mixtures were filtered through a glass Gooch crucible (porosity P2), under vacuum, washed with water and submitted to basic hydrolysis with 150 mL of potassium hydroxide (0.223 M), at boiling temperature and for 30 min. Subsequently, the mixtures were once more filtered through a glass Gooch crucible (porosity P2), under vacuum, washed with water and finally with acetone. The crucibles with the fibre were dried at 103 ± 2 °C (Laboratory heater, Memmert UL500, Schawabach, Germany) and weighted after cooling in a desiccator. Then, the sample was incinerated (3 h at 550 ± 25 °C, muffle furnace, Heraeus Hanau MR170E, Hanau, Germany), cooled in a desiccator, and reweighted. The total crude fibre content was expressed in g per 100 g wet weight.

#### 2.3.5. Protein

The protein content was determined by the Kjeldahl method [29]. Briefly, test portions of each sample were digested with concentrated sulphuric acid, in the presence of potassium sulphate and a low concentration of selenium catalyst, at 360 °C (Digestion System Tecator 2006, Höganäs, Sweden). During the digestion, nitrogen is released and retained as ammonium sulphate. After cooling to room temperature, ammonia was released from the acid digest by raising the pH with the addition of sodium hydroxide (6 M). Then, ammonia was distilled (Tecator Distilling Unit 1002, Höganäs, Sweden), collected in a boric acid solution, and titrated with a standardized sulphuric acid solution (0.02 N). Protein content was calculated using a conversion factor of 6.25 [30]. The analyses were performed in duplicate and the results were expressed in g per 100 g wet weight.

#### 2.3.6. Minerals

Minerals (B, Ca, Cu, Fe, K, Mg, Mn, P, S, Zn) in *Cajanus cajan*, *Lablab purpureus*, *Phaseolus vulgaris*, *Phaseolus lunatus* and *Vigna unguiculata* seeds were quantified by inductively coupled plasma optical emission spectrometry. Test portions of each sample were weighed and submitted to a digestion process with a mixture of nitric acid and hydrochloric acid (1:3, *v*/*v*) at 105 °C during 90 min and analysed using the Thermo Scientific iCAP 7000 Series ICP-OES spectrometer (Thermo Scientific, Cambridge, UK). Procedural blanks were prepared using the same analytical procedure and reagents. Calibration curves of, at least, five different concentrations were used to quantify each element. The analyses were performed in triplicate and the results expressed in mg per kg wet weight.

#### 2.3.7. Total Phenolic Content

Total phenolic compounds were determined according to Loebler et al. [31]. Briefly, water (6.0 mL), leaf ethanolic extract (100 μL) and undiluted Folin-Ciocalteu reagent (500 μL) were mixed in a 10.0 volumetric flask. After 1 min, 1500 μL of 20% (*w*/*v*) sodium carbonate was added and the volume was made up to 10.0 mL with water. After 2 h incubation at room temperature and in the dark, the absorbance was measured at 765 nm (SPEKOL 1500, Analytik Jena, Germany) and compared to a gallic acid calibration curve. The analyses were performed in triplicate and the results were expressed in mg of gallic acid equivalents per g of dry leaves.

#### 2.3.8. DPPH Radical Scavenging Capacity

The antioxidant capacity was determined according to the methodology described by Lima et al. [32]. Briefly, a 500 μL aliquot of diluted leaf ethanolic extract was added to 3 mL of daily prepared DPPH solution (24 mg/L in ethanol). After 30 min incubation at room temperature and in the dark, the absorbance was measured at 517 nm (SPEKOL 1500, Analytik Jena, Germany) and compared to an ascorbic acid calibration curve. The scavenging activity was measured as the decrease in absorbance of the samples versus DPPH standard solution. The analyses were performed in triplicate and the results were expressed in mg of ascorbic acid equivalents per g of dry leaves.

### 2.4. Agronomic Data Collection

The information on economic and agricultural profiles of each bean species under study was also investigated during the field surveys made between 2018 and 2019 at the main trade markets of Santiago Island. This information was complemented with data concerning the agro-economical characterization of Cabo Verde Islands, which was retrieved from: (i) agricultural data from the Ministry of Agriculture and Environment [33,34]; (ii) the National Institute of Statistics [27,35]; and (iii) the Annual Report of Cabo Verde [36,37].

### 2.5. Statistical Analyses

All data measurements are presented as mean values. Univariate analysis (UA) was performed to compare the chemical and nutritional traits among the bean species. Before running the UA, normality and homogeneity of variances were tested; as data did not follow normal distributions and the variances were not homogeneous, the test of means was carried out using Kruskal Wallis test for all variables (α = 0.05). After standardization (mean = 0, and standard deviation = 1) of chemical data, a multivariate analysis by principal component analysis (PCA), based on the correlation matrix, was performed and the eigenvectors and eigenvalues projected and visualized with the ggplot function of the ggplot2 package [38]. All analyses were carried out in the RStudio program version 1.1.456 [39].

## 3. Results

### 3.1. Diversity of Food Legume Species

Our results showed that 15 Leguminosae species were recognized as food plants in Cabo Verde and their native distribution, habit and distribution in Cabo Verde are presented in Table 1. Most of these species are non-native (73%, 11 species) and only four native species (27%) were accounted for. Also, 93% (14 species) of all species are cultivated, and *Zornia glochidiata* is the only one that is not so. Only 21% of the cultivated species (three species) are native to Cabo Verde. As previously mentioned, all 15 species are used for food, however, seven species (47%) are also described as medicinal, and eight species (53%) as forage and medicinal.

Considering the worldwide native distribution of each species (Table 1), four main groups were identified: Neotropical species (33%, five species) (e.g., *Arachis hypogaea*, *Phaseolus lunatus* and *Phaseolus vulgaris*); Oriental species (26.5%, four species) (e.g., *Cajanus cajan*, *Cassia fistula* and *Sesbania grandiflora*); Afrotropical species (26.5%, four species) (e.g., *Tamarindus indica*, *Lablab purpureus* and *Vigna unguiculata*); and Palearctic species (7%, one species: *Ceratonia siliqua*). Only one species, *Mucuna pruriens*, is distributed across two distinct biogeographical regions. The majority of the non-native food legume species used in Cabo Verde are from Neotropical and Oriental regions.

Morphologically, these 15 food legume species display a great diversity of habit. More than half (54%) are herbaceous, annual or biennial, 33% are trees, and only 13% correspond to shrubs (Table 1).

The majority of the studied food legumes (87%) are commonly found in Santiago (13 species), followed by Santo Antão (73%, 11 species) and Fogo (73%, 11 species); Boavista hosts the lowest number of species (13%, two species) (Figure 2; Table 1).

### 3.2. Chemical Composition

Ash, fibre, moisture and protein contents (%) of the five most cultivated and traded legume species in Cabo Verde were compared through boxplot analysis as shown in Figure 3. Overall, the contents did not differ considerably among these species (*p* < 0.05). However, mean ash contents of *Vigna unguiculata* (3.2 ± 0.3 g/100 g wet weight) were the lowest whereas *Phaseolus vulgaris* (4.1 ± 0.0 g/100 g wet weight) showed the highest ones. *Lablab purpureus* (7.8 ± 1.0 g/100 g wet weight) and *Cajanus cajan* (6.3 ± 0.1 g/100 g wet weight) were the richest species in fibre content and *Phaseolus lunatus* and *Phaseolus vulgaris* the poorest (3.7 ± 1.0 g/100 g wet weight and 3.7 ± 0.4 g/100 g wet weight, respectively). The average moisture contents ranged from 10.7 ± 0.2 g/100 g (*Phaseolus vulgaris*) to 12.4 ± 0.9 g/100 g (*Phaseolus lunatus*). The highest contents in protein were measured in *Lablab purpureus* (23.3 ± 0.6 g/100 g wet weight), followed by *Phaseolus vulgaris* (23.2 ± 0.4 g/100 g wet weight), *Vigna unguiculata* (23.0 ± 1.4 g/100 g wet weight), *Cajanus cajan* (22.0 ± 2.0 g/100 g wet weight) and *Phaseolus lunatus* (19.0 ± 1.8 g/100 g wet weight).

The highest leaf phenolic contents (Table 2) were found in *Cajanus cajan* (4.55 ± 0.16 mg AGE/mg dry weight) and *Lablab purpureus* (4.13 ± 0.28 mg AGE/mg dry weight). On the other hand, the lowest contents were 3.15 ± 0.48 mg GAE/mg dry weight (*Phaseolus lunatus*), 3.04 ± 0.15 mg GAE/mg dry weight (*Vigna unguiculata*) and 3.02 ± 0.08 mg GAE/mg dry weight (*Phaseolus vulgaris*). Consistent with the phenolic contents, the antioxidant capacities of *Cajanus cajan* and *Lablab purpureus* were the highest (Table 3), respectively, 2.35 ± 0.08 mg AAE/mg dry weight and 2.49 ± 0.11 mg AAE/mg dry weight. Accordingly, *Phaseolus lunatus* (1.75 ± 0.44 mg AAE/mg dry weight), *Vigna unguiculata* (1.84 ± 0.11 mg AAE/mg dry weight), and *Phaseolus vulgaris* (2.22 ± 0.05 mg AAE/mg dry weight) presented the lowest antioxidant capacities.

Table 3 presents the mineral contents of the five most cultivated and traded beans of Cabo Verde: *Cajanus cajan*, *Lablab purpureus*, *Phaseolus lunatus*, *Phaseolus vulgaris* and *Vigna unguiculata*. The B contents ranged from 5.8 ± 0.6 mg/kg wet weight (*Lablab purpureus*) to 9.2 ± 1.8 mg/kg wet weight (*Vigna unguiculata*). The average contents of Ca ranged from 581.9 ± 97.8 (*Lablab purpureus*) to 1418.5 ± 63.3 mg/kg wet weight (*Phaseolus vulgaris*). The Cu contents ranged between 4.8 ± 0.7 (*Vigna unguiculata*) and 9.4 ± 2.4 mg/kg wet weight (*Cajanus cajan*). Fe values varied between 39.6 ± 1.3 mg/kg wet weight (*Cajanus cajan*) and 86.5 ± 21.0 mg/kg wet weight (*Phaseolus lunatus*). K contents ranged between 7607.5 ± 599.1 mg/kg wet weight (*Phaseolus lunatus*) and 11,704.7 ± 201.1 (*Phaseolus vulgaris*). On average, *Cajanus cajan* has the lowest Mg content (1229.2 ± 35.5 mg/kg wet weight) and *Vigna unguiculata* the highest (1899.3 ± 123.4 mg/kg wet weight). *Cajanus cajan* showed the lowest Mn content (15.7 ± 0.3 mg/kg wet weight) and *Lablab purpureus* the highest (26.4 ± 1.5 mg/kg wet weight). The lowest P content was 3662.4 ± 335.2 mg/kg wet weight (*Cajanus cajan*) and the highest was 4370.3 ± 181.1 mg/kg wet weight (*Phaseolus vulgaris*). The S content of *Phaseolus lunatus* was the lowest (1483.4 ± 326.1 mg/kg wet weight) while *Vigna unguiculata* exhibited the highest (1937.2 ± 46.0 mg/kg wet weight). Finally, *Phaseolus vulgaris* showed a Zn content of 21.7 ± 1.8 mg/kg wet weight and *Vigna unguiculata* a mean value of 27.2 ± 3.5 mg/kg wet weight, respectively the highest and lowest values.

In order to assess the patterns of variations in the mineral contents of the five Cabo Verde bean species (Table 3), a Principal Components Analysis (PCA) was performed (Figure 4). The first four principal components (PCs) accounted for 77.40% of the variability amongst the five species (see Appendix A). The first PC (PC1) accounted for 29.36% of the total variation with B, K and P presenting negative coefficients. PC2 accounted for additional 20.76% of the total variation, with Ca and Mn being the most important contributors, the first with a negative coefficient and the latter with a positive one. PC3 accounted for further 16.58% of the variability and showed Cu as a strong negative contributor and Mg as a positive one. PC4 accounted for 10.70% of the variability, describing the patterns of variation of Fe and Zn with positive coefficients, and S with negative one.

The differentiation patterns between the five species, for PC1 and PC2, are presented in Figure 4, showing a large mineral diversity that slightly distinguishes the species. *Lablab purpureus* showed positive values of both components while *Phaseolus vulgaris* showed negative values. *Cajanus cajan* displayed positive values for PC1 and negative values for PC2, contrasting with *Vigna unguiculata* which showed positive values for PC2 and negative values for PC1. Finally, *Phaseolus lunatus* presented a broad variety of mineral contents with both negative and positive values of PC1 and PC2.

### 3.3. Agronomic Analysis

Figure 5 shows that the evolution of the bean cropland area over the last decade followed a trend similar to that of rainfed crops. The bean cropland in Cabo Verde represents about 50% of the total extant rainfed cropland. In 2007, 2010, 2013 and 2015 the overall cultivated area decreased. Over the studied period, the maximum cultivated area was reached in 2008, corresponding to 66,411 ha in the whole and to 34,385 ha in bean cropland.

According to Figure 6, the annual production of beans as traditional rainfed crops severely fluctuated from 2013 to 2017, decreasing from 5943 to 700 tons in 2014 and increasing to 5199 tons in 2015. The lowest bean production happened in 2017, and represented a drastic reduction of 99.8% when compared with 2016, while in 2012 the maximum was observed (5950 tons), over the studied period. Overall, a negative linear trend of beans production was observed from 2006 to 2017.

## 4. Discussion

### 4.1. Food Legume Species in Cabo Verde

Legumes supply nutrients and less expensive non-animal proteins to meet the needs of people, particularly of Low- and Middle- Income Countries, and surpluses can be sold to generate family income [40]. The present study allowed us to identify five pulses (*Cajanus cajan*, *Lablab purpureus*, *Phaseolus lunatus*, *Phaseolus vulgaris* and *Vigna unguiculata*) that are widely traded in Santiago markets, but also to identify the Leguminosae species used as food in Cabo Verde. Fifteen legume species are used as food, with a significant share described as medicinal (47%) and as forage (53%). All of those species are cultivated, except for *Zornia glochidiata*, which is common in Sudano-sahelian pastures [41]. There is a predominance of species from Tropical regions, the places of origin and domestication of a large number of the species cultivated in the archipelago, like beans (*Canavalia ensiformis*, *Lablab purpureus*, *Phaseolus lunatus*), peanuts (*Arachis hypogaea*) or manioc (*Manihot esculenta*: family Euphorbiaceae) [16]. Presently, *Cajanus cajan*, *Vigna unguiculata*, *Lablab purpureus*, *Phaseolus lunatus* and *Phaseolus vulgaris* continue to have significant importance at both food and medicinal levels, as reported in previous studies [10,15].

### 4.2. Chemical Composition

Legumes are known to significantly contribute to the supply of bioactive compounds to the body due to their antioxidant activity, attributed to phenolic compounds, and are also rich sources of proteins, dietary fibres, and micronutrients. Legumes have also acquired a significant importance in African traditional medicine [42,43]. Our results revealed that *Lablab purpureus*, *Phaseolus vulgaris* and *Vigna unguiculata* have higher protein contents (~23%), with the former containing also high levels of phenolic contents and antioxidant capacities in its leaves, together with *Cajanus cajan*. Despite the high antioxidant and phenolic contents of legume leaves, they are currently used to feed animals during fodder shortages in the dry season. Seeds, as part of daily diet, are the most consumed plant parts in Cabo Verde [44]. Neglecting legume leaves as food has been more frequent in Cabo Verde than in other West African countries, where they are commonly cooked in stews [45]. In a country where food shortage and malnutrition still prevails, such as Cabo Verde, the promotion of leaves in the diet should be considered due to high health benefits as anticarcinogenic, anti-inflammatory, antioxidant, and anti-microbial (e.g., [46]).

Our results revealed that ash contents ranged from 3.2% (*Vigna unguiculata*) to 4.1% (*Phaseolus vulgaris*), indicating that they are good sources of minerals (e.g., [47,48]). Moreover, all the studied species revealed an average moisture content ranging from 10.7 ± 0.2 g/100 g in *Phaseolus vulgaris* to 12.4 ± 0.9 g/100 g in *Phaseolus lunatus* and we concluded that it is possible to store all these beans with quality, in agreement with other studies (e.g., [49]). Amarteifio et al. [50] recorded fibre contents of *Cajanus cajan* cultivated in Botswana ranging from 9.8 to 13.0 g/100 g, higher than those we obtained (6.3 g/100 g). This difference can be explained by the different provenances of the samples, taking the particular environmental aspects of each region into account.

The presence of iron and zinc in the screened beans species is vital, as these micronutrients are responsible for essential body functions, and a deficiency in these minerals can lead to severe medical conditions [51]. Iron is needed for the transfer of oxygen to body tissues and organs; it is the most common nutrient deficiency, affecting over 2 billion people worldwide, and a major public health burden concerning African children [52]. Zinc plays an essential role in body metabolism and it prevents illnesses by supporting the immune system [53]. The zinc and iron contents show significant differences between bean species, with *Phaseolus vulgaris* and *Phaseolus lunatus* being the iron-richest species, but *Phaseolus vulgaris* having a lower zinc content. These results highlight a species-specific profile of zinc and iron contents, which does not significantly affect the nutritional value of each bean species. A study conducted in Cabo Verde by Semedo et al. [54] revealed a prevalence of anaemia of 51.8%, particularly high in children below 24 months of age living under poor household conditions, thus highlighting anaemia as a public-health concern in the country. In addition to the traditional use of beans as a multipurpose crop, the increase of its consumption at household level could be foreseen to overcome some dietary deficiencies (e.g., iron).

### 4.3. Agronomic Value of Pulses

The development of Cabo Verde rural economy in recent years has been negative due to three consecutive years of drought, with irregular and insufficient rainfall [55]. Included in the Sahel region, the archipelago experiences periods of drought characterized by the absence of rain, low rainfall or by its poor distribution over the rainy season [5]. Natural obstacles have historically been a challenge to Cabo Verde’s rural development. The contribution of agriculture to the rural economy decisively obeys to the favourable or adverse climatic conditions [16].

Our study showed that the evolution of the bean cultivated area follows a trend similar to the other rainfed crops, a consequence of being established alongside the maize crops (e.g., [56]). The cultivation of the five bean species under study is of great importance, in terms of consumption/trade, for the food of rural households, occupying most of the agricultural areas of Santiago Island and dictating the food culture of this population [57]. Despite the decreasing trend of cultivated areas in Cabo Verde, rainfed farming occupies a large portion, with 89.2% of the agricultural area occupied by maize and beans [34].

The different bean species are traditionally cultivated in association with maize, except for *Cajanus cajan* which is sown randomly [58]. This is a seasonal crop, directly dependent on the few months of rain (two to three months per year) concentrated between August to October [59]. According to Temudo [60], *Phaseolus vulgaris* is the most climatically demanding and the only bean species grown in the higher and cooler zones, where the precipitation is more regular and the air humidity is higher. On the other hand, *Lablab purpureus* is the most drought-resistant bean species [60].

Altogether, the total production of traditional rainfed crops fluctuated from 2013 to 2017, decreasing from 12,008 (2013) to 700 tons (2014) and increasing to 9739 tons in 2016 [33]. This variation pattern is also reflected in our results for beans, with an evident and strongly random character determined by the meteorological effects on crop production. It is worth noting that the negative trend of bean production contrasts with the average annual growth population rate of 12.1% over the last 10 years (2006–2016) [27]. Moreover, it must be pointed out that *Cajanus cajan* and *Lablab purpureus* accounted for more than 50% of the total bean production [33].

The rainfed production in Cabo Verde still does not meet the needs of the population, ensuring only 10–15% of the national food consumption, thus forcing the huge importation of food supplies [61]. This also applies to beans, as in addition to local production, a considerable consumption depends on importation of both canned and dried beans, mainly in the urban areas, throughout the year, whereas the local production is consumed in the rural areas during a restricted period. Generally, the cultivated bean species are primarily used for household self-support; however, in years of good harvests, beans are sold in the local markets [34].

### 4.4. Food Security and Pulses in Cabo Verde

Cabo Verde’s history records cyclical famines, of which some stand out in the 1920s, 1940s and 1970s, which decimated thousands of people [62]. More recently, Cabo Verde reported almost no harvests for the 2017–2018 agricultural season due to a severe drought, highlighting that 28,000 people (5.3%) are currently facing food insecurity (Data available at http://www.west-africa-brief.org/content/en/partners-help-cabo-verde-cope-food-insecurity). Besides the environmental challenges to agriculture productivity, a recent study [4] revealed that Cabo Verde food production still falls short on meeting internal consumers’ needs, posing a huge threat to food security and a high dependence on food imports [63]. Despite the considerable progress made since independence (in 1975) in fighting poverty, Cabo Verde has not yet eradicated hunger and, according to FAO, about 20% of rural households suffer from food insecurity [64].

Therefore, the lack of rain for the production of maize and beans leads to profound changes in the Cabo Verde food pattern: according to Silva [44], the consumption of traditional products (maize and beans) that are the basis of the diet in Cabo Verde is being replaced by the consumption of 100% imported rice. This decrease in the production of maize and beans consequently leads to a decrease in grain consumption. Moreover, the purchase of food becomes more expensive. In this way, the food and nutritional security of families is negatively affected, by reduction in the number of daily calories needed to keep their dietary needs balanced.

Grain legumes are a very important food crop in many parts of Africa, as they are a source of high-protein products, and have the advantage of fixing atmospheric nitrogen which enriches the soil thereby reducing the cost of fertilizer inputs, especially in nutrient poor soils [65]. In Cabo Verde, rainfall scarcity and soil infertility, together with the arid territory pose a challenging scenario for agriculture production [16]. It should be noted that the consumption of beans dates back to the first settlements of these islands, with the broad bean (*Vicia faba*) introduced by the Portuguese settlers and the African beans being brought to Cabo Verde from the western coasts of Africa, and from the New World, along with the slave trade [66].

Maize has been a staple crop in Cabo Verde since its introduction from the Americas and still today, followed by legumes used for both food and fodder. Most of the beans traded in Santiago seem to be cultivated in Cabo Verde for several years, intercropping with potatoes and maize. Bean varieties and/or landraces adapted to Cabo Verde’s severe conditions have been selected at smallholders’ level. *Vigna unguiculata* (cowpea) is a drought-tolerant food crop, well adapted to a diverse range of climate and soil types, and widely cultivated throughout the tropics and subtropics [67,68]. In Africa, cowpea is mainly cultivated in West and Central Africa, with an annual production of 3 million tons, being also known as the poor man’s meat [69]. Also, *Lablab purpureus* (hyacinth bean) is among the so-called ‘lost crops’ but with potential to become an important crop species in the future due to its enhanced environmental tolerances when compared to other legumes [70]. Considering its inherent environmental resilience, it would be important to conduct future studies on chemical and genetic diversity focused on these native bean species, which are adapted to extreme climatic conditions. For instance, *Lablab purpureus*, which occurs in dry coastal areas in Santiago Island, while *Cajanus cajan* and *Phaseolus vulgaris* are cultivated mostly at humid and elevation zones, which may indicate that the cultivation of these beans by the rural communities considers the adaptation to adverse habitats for each bean species. Former studies in plant genetic resources from Cabo Verde highlighted the predominance of several important Crop Wild Relatives (CWR) taxa occurring under extreme habitat conditions that are well adapted to drylands and poor soils, namely in Brassicaceae CWR (*Diplotaxis*, [71]) and in sugar beet CWR (*Patellifolia procumbens* [72]), and an important center of CWR diversity of West African crop millets [73]. These studies underline the high predominance of agrobiodiversity hotspots associated with abiotic stress adaptation (i.e., drought and salinity), by selecting places of origin where they are cultivated, in order to determine any climatic influence on chemical composition and evaluate the potential of Cabo Verdean bean species adapted to the extreme conditions of the country.

Since the historical cultivation of the native bean species under the extreme environmental conditions of Cabo Verde, ex situ conservation measures should be considered to conserve these invaluable resources, as only 17 bean accessions (Appendix A) from Cabo Verde are held at germplasms banks [74], namely: 14 of *Cajanus cajan*, two of *Phaseolus* (*Phaseolus lunatus* and *Phaseolus vulgaris*), and one of *Vigna unguiculata*. The adverse climatic conditions of this archipelago, with cyclical drought periods, surely drove acclimation processes and probably multiple landraces were locally developed in these islands. Interestingly, and despite the small number of edible grain legumes, more than 50% of the germplasm accessions reported from Cabo Verde are forage legumes (namely from the endemic-rich genus *Lotus*), revealing the interest of this family for pasture improvement [22]. Ex situ conservation of the plant genetic resources in Cabo Verde, especially concerning food species commonly used by the population, has been very limited, with few efforts to characterize, evaluate, and preserve this genetic heritage. Besides its great importance at nutritional and agronomic levels, the income raised from legume sales significantly contributes to food security at the household level [75].

Legumes are a good source of protein and micronutrients and could hence be a good complement to starchy diets, where deficiency of protein is a concern [76]. Protein contents in legumes frequently range from 20 to 45%, which means a higher protein content than most plant foods and twice the protein content of cereals [77,78].

## 5. Conclusions

The present study involved an extensive research about the diversity of cultivated food legume species in Cabo Verde, focusing on the most consumed/traded pulses: *Cajanus cajan*, *Lablab purpureus*, *Phaseolus lunatus*, *Phaseolus vulgaris*, and *Vigna unguiculata*. Through multiple approaches—legumes diversity and local uses; chemical, nutritional and antioxidant evaluation; complemented with agro-economic analysis—we discuss the results in the light of conservation and sustainable use of these legumes, and their potential contribution for food security in this archipelago. Our results revealed that the studied bean species are excellent sources of minerals, proteins, phenols, and antioxidants, representing an invaluable potential to satisfy the nutritional needs of Cabo Verde populations. Beans represent about half of Cabo Verde’s crop production which, together with maize, corresponds to about 90% of the total cropland. However, the adverse climatic conditions of this archipelago, characterised by the scarcity or irregularity of rainfall, causes a drastic decrease in local crop production. This translates into lack of food products, the need to import and the increase of prices, worsening the nutritional insecurity by reducing access to food. Under this scenario of food and socio-economic crisis, new strategies and investments must be implemented, such as the use of drought-resistant cultivars, new agricultural production techniques, and water saving and desalination systems, since most of the local population largely depends on domestic crop productivity for self-support.

Finally, our study highlights a limitation in ex situ conservation of the plant genetic resources of Cabo Verde. Thus, further field surveys are still needed, involving new efforts to enhance both in situ and ex situ conservation of these species, specifically to assess, catalogue, and preserve their genetic legacy that can be used in bean breeding programs.

## Figures and Tables

**Figure 1 foods-10-00206-f001:**
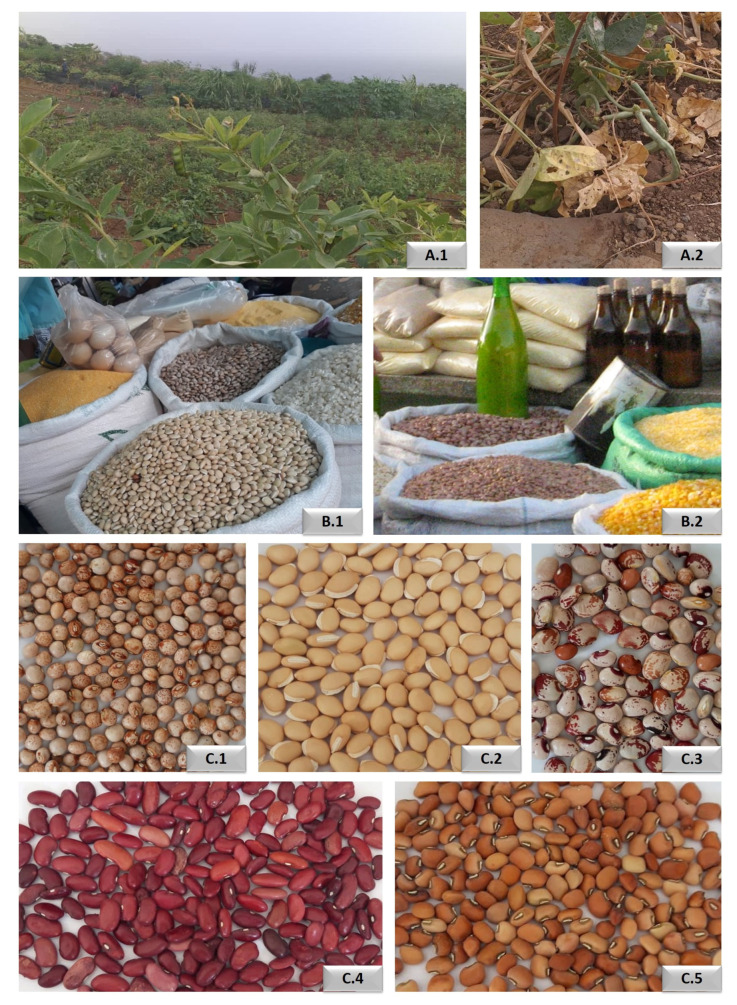
Studied bean species from Santiago Island. (**A**) Field beans in the Assomada region, *Cajanus cajan* (**A.1**) and *Vigna unguiculata* (**A.2**). (**B**) The two main Santiago markets showing different beans sold in Assomada market (**B.1**) and Cidade da Praia municipal market (**B.2**). **C.** Seeds of the studied bean species: *Cajanus cajan* (**C.1**); *Lablab purpureus* (**C.2**); *Phaseolus lunatus* (**C.3**); *Phaseolus vulgaris* (**C.4**); and *Vigna unguiculata* (**C.5**).

**Figure 2 foods-10-00206-f002:**
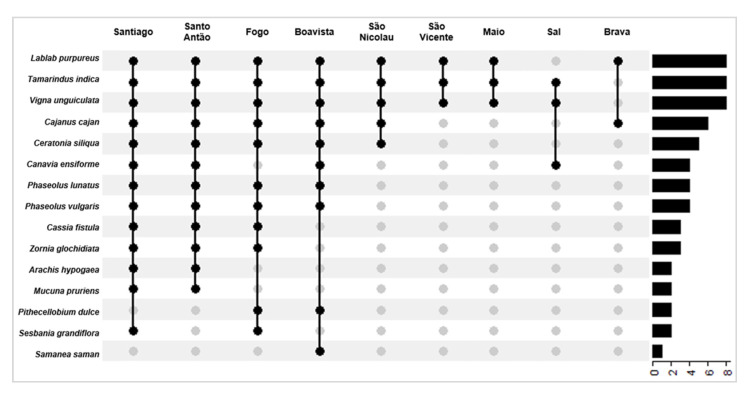
Distribution of the food legume species in Cabo Verde. UpSet diagram showing the presence and number of species per island. The lines linking the dots represent species that occur in two or more islands.

**Figure 3 foods-10-00206-f003:**
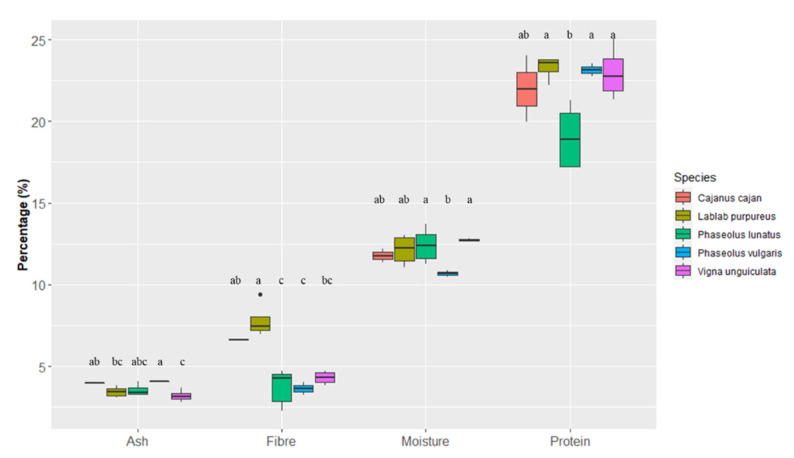
Intraspecific and interspecific variation of ash, fibre, moisture and protein contents in the beans of the five most cultivated and traded legume species of Cabo Verde. The box represents the 25th, 50th (median) and 75th percentiles, while whiskers represent the 10th and 90th percentiles with minimum and maximum observations. The black dots represent the outliers. Species sharing one or more letters for each trait are not statistically different (*p* < 0.05).

**Figure 4 foods-10-00206-f004:**
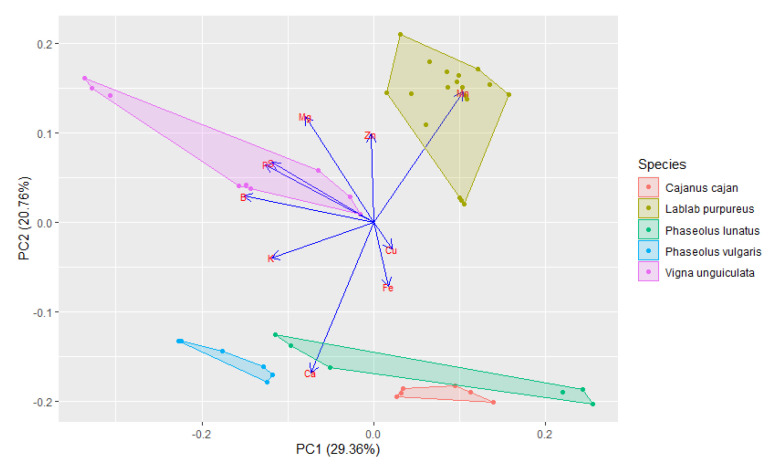
Representation of the two first components (PC1 and PC2) resulting from the principal component analysis, which explain 50.12% of the diversity of mineral contents in beans of the five most cultivated and traded legume species of Cabo Verde in a space defined by the vectors and own values. The arrow lengths show differences in variance explained relative to each other.

**Figure 5 foods-10-00206-f005:**
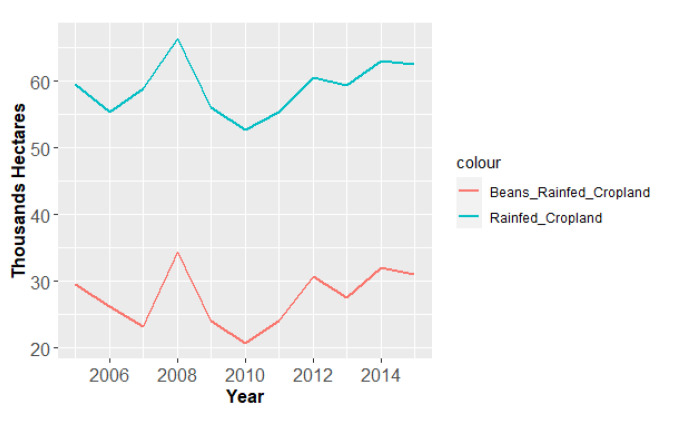
Areas of extant rainfed cropland and beans rainfed cropland in Cabo Verde, from 2005 to 2015 [33,35].

**Figure 6 foods-10-00206-f006:**
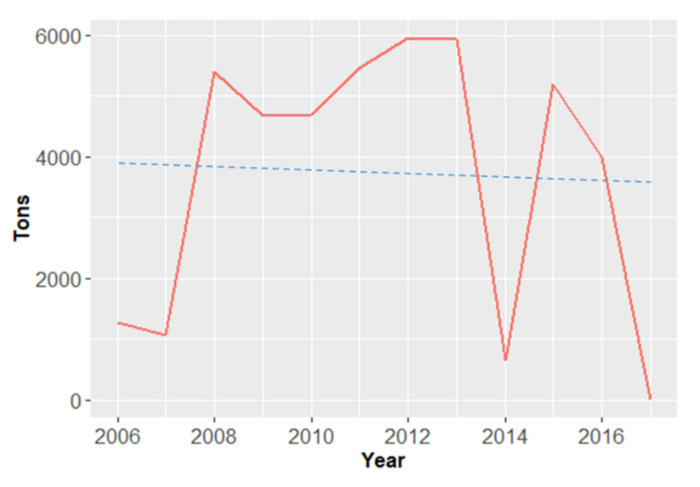
Annual bean production (in tons) of the five most cultivated and traded legume species in Cabo Verde, from 2006 to 2017 [27,33].

**Table 1 foods-10-00206-t001:** Food legume species in Cabo Verde: scientific name, common name, status, important uses in Cabo Verde, native distribution, and distribution in Cabo Verde.

Species	Common Names	Status ^a^	Important Uses	Habit	Native Distribution	Distribution in Cabo Verde ^b^
*Arachis hypogaea* L.	Groundnut, Peanut	Ic	Food, Forage, Medicinal	Annual herb	Neotropical	A, T
*Cajanus cajan* (L.) Millsp.	Pigeon pea	Ic	Food, Forage, Medicinal	Shrub	Oriental	A, N, B, T, F, Br
*Canavalia ensiformis* L.	Jack bean	Ic	Food, Medicinal	Annual herb	Oriental	A, M, T, Br
*Cassia fistula* L.	Golden shower	Ic	Food, Medicinal	Tree	Oriental	A, T, F
*Ceratonia siliqua* L.	Carob tree	Ic	Food, Forage, Medicinal	Tree	Palearctic	A, N, T, F, Br
*Lablab purpureus* (L.) Sweet	Hyacinth bean	Nc	Food, Forage, Medicinal	Annual herb	Afrotropical	A, V, N, S, B, T, F, Br
*Mucuna pruriens* (L.) DC.	Velvet bean	Nc	Food, Forage, Medicinal	Annual herb	Afrotropical-Oriental	A, T
*Phaseolus lunatus* L.	Lima bean	Ic	Food, Medicinal	Annual or biennial herb	Neotropical	A, T, F, Br
*Phaseolus vulgaris* L.	Common bean, Kidney bean	Ic	Food, Medicinal	Annual or biennial herb	Neotropical	A, T, F, Br
*Pithecellobium dulce* (Roxb.) Benth.	Manila tamarind	Ic	Food, Medicinal	Tree	Neotropical	F, Br
*Samanea saman* (Jacq.) Merr.	Monkeypod	Ic	Food, Forage, Medicinal	Tree	Neotropical	Br
*Sesbania grandiflora* (L.) Pers.	Agati sesbania	Ic	Food, Medicinal	Shrub	Oriental	T, F
*Tamarindus indica* L.	Tamarind	Ic	Food, Forage, Medicinal	Tree	Afrotropical	A, V, N, S, M, T, F, Br
*Vigna unguiculata* (L.) Walp.	Cowpea	Nc	Food, Forage, Medicinal	Annual herb	Afrotropical	A, V, N, S, M, T, F, Br
*Zornia glochidiata* Reichb. ex DC.	Herbe mouton	N	Food, Medicinal	Annual herb	Afrotropical	A, T, F

**^a^****I**, Introduced; **N**, Native; **c,** cultivated; **^b^**
**Distribution in Cabo Verde**: Islands: **A**, Santo Antão; **V**, São Vicente; **N**, São Nicolau; **S**, Sal; **B**, Boavista; **M**, Maio; **T**, Santiago; **F**, Fogo; **Br**, Brava.

**Table 2 foods-10-00206-t002:** Mean values, standard deviations and homogeneous groups ^1^ of the antioxidant capacities of the five most cultivated and traded legumes of Cabo Verde.

	*Cajanus cajan*	*Lablab purpureus*	*Phaseolus lunatus*	*Phaseolus vulgaris*	*Vigna unguiculata*
**Total Phenolic Content** (mg GAE/mg dry weight) ^2^	4.55 ± 0.16 a	4.13 ± 0.28 a	3.15 ± 0.48 b	3.02 ± 0.08 b	3.04 ± 0.15 b
**DPPH radical scavenging capacity** (mg AAE/mg dry weight) ^3^	2.35 ± 0.08 a	2.49 ± 0.11 a	1.75 ± 0.44 b	2.22 ± 0.05 ab	1.84 ± 0.11 b

^1^ Homogeneous groups: species sharing one or more letters for each variable are not statistically different (*p* < 0.05). ^2^ GAE, Gallic acid equivalents. ^3^ AAE, Ascorbic acid equivalents.

**Table 3 foods-10-00206-t003:** Mean values (mg/kg wet weight), standard deviations and homogeneous groups ^1^ for the mineral contents in the beans of the five most cultivated and traded legume species of Cabo Verde.

Minerals	*Cajanus cajan*	*Lablab purpureus*	*Phaseolus lunatus*	*Phaseolus vulgaris*	*Vigna unguiculata*
**B**	6.2 ± 0.3 bc	5.8 ± 0.6 c	6.1 ± 1.1 bc	7.0 ± 1.2 b	9.2 ± 1.8 a
**Ca**	1144.4 ± 67.0 a	581.9 ± 97.8 c	1332.0 ± 438.5 a	1418.5 ± 63.3 a	818.1 ± 72.2 b
**Cu**	9.4 ± 2.4 a	6.8 ± 2.5 ab	6.0 ± 1.4 bc	6.6 ± 0.6 ab	4.8 ± 0.7 c
**Fe**	39.6 ± 1.3 c	54.4 ± 3.1 b	86.5 ± 21.0 a	76.0 ± 3.6 a	54.0 ± 2.0 b
**K**	9148.6 ± 1466.8 bc	8255.3 ± 1314.5 cd	7607.5 ± 599.1 d	11,704.7 ± 201.1 a	10,149.0 ± 282.6 b
**Mg**	1229.2 ± 35.5 b	1820.7 ± 73.4 a	1726.0 ± 230.6 a	1798.4 ± 43.7 a	1899.3 ± 123.4 a
**Mn**	15.7 ± 0.3 c	26.4 ± 1.5 a	18.9 ± 0.9 b	16.2 ± 1.7 c	17.1 ± 1.7 bc
**P**	3662.4 ± 335.2 b	4004.5 ± 151.0 b	3782.8 ± 267.5 b	4370.3 ± 181.1 a	4167.4 ± 728.1 b
**S**	1555.3 ± 61.0 c	1689.9 ± 168.7 b	1483.4 ± 326.1 bc	1859.8 ± 54.0 a	1937.2 ± 46.0 a
**Zn**	25.6 ± 0.5 a	26.0 ± 5.0 a	22.9 ± 1.3 b	21.7 ± 1.8 b	27.2 ± 3.5 a

^1^ Homogeneous groups: species sharing one or more letters for each mineral are not statistically different (*p* < 0.05).

## Data Availability

Data is contained within the article or Appendix A.

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
