# Peer review of "Tackling Food Insecurity in Cabo Verde Islands: The Nutritional, Agricultural and Environmental Values of the Legume Species"

_foods, 2021, doi:10.3390/foods10020206_

Round 1

Reviewer 1 Report

Reviewed manuscript entitled „Tackling food insecurity in Cabo Verde Islands: the nutritional, agricultural and environmental values of the legume species” is an original research study.

Following are the detailed suggestions for the Authors:

Materials and Methods

What varieties of legumes were studied?

At what stage of plant development the leaves were collected for analysis? Whether the leaves were fully developed and mature.

What were the conditions for growing legumes in a greenhouse (soil, watering, temperaturÄ™, lighting, humidity)?

What was the sample size for chemical analyzes?

line 127, should be „H2SO4

line 147, should be „25ºC” - please correct it throughout

line 235, should be „Na2CO3

line 236, should be „H2O”

Results

line 409, should be „In 2007, 2010, 2013 and 2015 the overall cultivated…”

Discussion

remove „e.g.” form: line 449, „(e.g., [16])” - please correct it throughout

line 460 „Cajanus cajan” should be in italic

Author Response

REVIEWER#1

COMMENTS AND SUGGESTIONS FOR AUTHORS

Reviewed manuscript entitled “Tackling food insecurity in Cabo Verde Islands: the nutritional, agricultural and environmental values of the legume species” is an original research study. Following are the detailed suggestions for the Authors:

RESPONSE: Thanks to the reviewer’s questions, we have realized that there was some misleading information in the former version of the manuscript and we have revised the new version accordingly.

MATERIALS AND METHODS

What varieties of legumes were studied?

RESPONSE: Although we agree with the Reviewer’s concern, we would like to state that our study was focused only on beans marketed and consumed in Santiago Island. The five bean species studied have different varieties, but only Lablab purpureus, native in Cabo Verde, has two distinct varieties that are commonly sold in the Santiago Island markets (i.e.: var. albiflorus, which includes those beans with white flowers and white or rusty seeds; and var. purpureus, which includes purple flowers and black seeds). We would like to emphasize that we have done a preliminary survey and the results revealed that no significant differences were found between these varieties and since our sampling was restricted to the Santiago markets, we prefer to focus our study on differences among the 5 species. However, in the revised version of our manuscript we provide more details on Supplementary Figure 1, as also suggested by the Reviewer 2.

At what stage of plant development the leaves were collected for analysis? Whether the leaves were fully developed and mature.

RESPONSE: We would like to state that the leaves collected in order to obtain leaves needed to further evaluate in the laboratory their antioxidant capacity were developed and mature, namely collected after 3 weeks in the case of Phaseolus vulgaris and to 2 month in the case of Cajanus cajan. This is now provided in the new version (Line 161)

What were the conditions for growing legumes in a greenhouse (soil, watering, temperature, lighting, humidity)?

RESPONSE: All the Leguminosae species were cultivate in greenhouse at Instituto Superior de Agronomia of the University of Lisbon (ISA/UL), in order to obtain leaves needed to further evaluate in the laboratory their antioxidant capacity. All the species were cultivated in pots with fertile and well-drained soils with the optimal temperature for germination varying between 25°C and 27°C. All the seeds germinated 10 – 20 days after planting. This information is now provided in the new version (Lines 162-164)

What was the sample size for chemical analyzes?

RESPONSE: Thanks for the comments. For chemical analyzes we used 100 g seeds of each bean accession. This information is now provided in the new version (Line 225). Specifically, for each of the analyzes, a different number of replicates was applied as we can see: line 240: "analyzes were performed in duplicate" (for ash and moisture); we added in line 243: "in duplicates" (for crude fiber); lines 266 "performed in duplicate" (for protein); line 277 "performed in triplicate" (for minerals).

line 127, should be H2SO4

RESPONSE: Thanks for the comments. We changed to: “sulfuric acid” as suggested by the Reviewer 2 (line 245).

line 147, should be „25ºC” - please correct it throughout

RESPONSE: Thanks for the comments, it has been revised

line 235, should be Na2CO3

RESPONSE: Thanks for the comments. We changed to: “sodium carbonate” as suggested by the Reviewer 2 (line 282-283).

line 236, should be H2O

RESPONSE: Thanks for the comments, it has been revised

RESULTS

line 409, should be „In 2007, 2010, 2013 and 2015 the overall cultivated…”

RESPONSE: Thanks for the comments, it has been revised (lines 458-459)

DISCUSSION

remove „e.g.” form: line 449, „(e.g., [16])” - please correct it throughout

RESPONSE: Thanks for the comments, it has been revised

line 460 Cajanus cajan should be in italic

RESPONSE: Thanks for the comments, it has been revised (line 509)

Reviewer 2 Report

A very well written article that address the current challenge in the food supply chain in a timely manner. In particular, the article provides useful insights on the utilisation of legumes. Though the article focused merely on a specific geographical location, the information summarised in this article would be useful for many readers facing the same challenge. Listed below are a few minor suggestions to improve the manuscript:

  • L48. Full name for FAO
  • L63. Full name for IUCN
  • L99. Full name for N and NE
  • L158. Full name for ISA and UL
  • L168. Full name for ICP
  • L179. Please provide the model and manufacturer name of the grinder
  • L181. Please specify the duration of drying and at what temperature. Is it air drying?
  • L197. Full name for H2SO4
  • L198 and L200. Please specify the boiling temperature
  • L200. Full name for KOH
  • L223. Please provide the model and manufacturer name for the ICP-MS
  • L235. Full name for Na2CO3 and H2O
  • L247. Please justify why ascorbic acid calibration curve was utilised to represent the DPPH result
  • Section 2.4. It would be useful if the authors could find the weather data (e.g. temperature, humidity, rainfall, sunshine hours, soil depth and moisture....) for the different places on the island in the last 3-5 years to identify if weather also play a role in influencing their chemical composition. Please add the information if available. Could be useful for the discussion section as a strong supporting evidence.
  • L254. Please clarify the term "expert-wise approach"
  • Fig 1 and Table 1. Please provide the images for the legumes and their common name.
  • Figure S1. Are these different places on the same altitude? Please add the information if available.

Author Response

REVIEWER#2

A very well written article that address the current challenge in the food supply chain in a timely manner. In particular, the article provides useful insights on the utilisation of legumes. Though the article focused merely on a specific geographical location, the information summarised in this article would be useful for many readers facing the same challenge. Listed below are a few minor suggestions to improve the manuscript.

RESPONSE: Thank you very much for this positive feedback. We appreciate very much the constructive and helpful comments of the reviewer. We have changed the manuscript to solve these shortcomings. Please see our detailed responses below.

L48. Full name for FAO

RESPONSE: Thanks for the comments, it has been revised. We changed to: “Food and Agriculture Organization (FAO)”.

L63. Full name for IUCN

RESPONSE: Thanks for the comments, it has been revised. We added: “International Union for Conservation of Nature (IUCN)” (line 64).

L99. Full name for N and NE

RESPONSE: Thanks for the comments, it has been revised. We changed to: “North and Northeast” (line 101).

L158. Full name for ISA and UL

RESPONSE: Thanks for the comments, it has been revised. As mentioned in the text ISA/UL is an abbreviation of “Instituto Superior de Agronomia of the University of Lisbon” (line 160).

L168. Full name for ICP

RESPONSE: Thanks for the comments, it has been revised. We added: “inductively coupled plasma optical emission spectrometry (ICP-OES)” (lines 214-215).

L179. Please provide the model and manufacturer name of the grinder

RESPONSE: Thanks for the comments, it has been revised. A common “Kunft” coffee mill” was used. The information was added to the manuscript (line 226).

L181. Please specify the duration of drying and at what temperature. Is it air drying?

RESPONSE: Thanks for the comments. The leaves were dried for two weeks, in the dark, at room temperature. The information was added to the manuscript (lines 228-229).

L197. Full name for H2SO4

RESPONSE: Thanks for the comments, it has been revised. We changed to: “sulfuric acid”.

L198 and L200. Please specify the boiling temperature

RESPONSE: Thanks for the comment. The standard does not specify the temperature but only indicates that the samples should be heated until they boil and that they should be kept boiling for 30 minutes. That was exactly what was done.

L200. Full name for KOH

RESPONSE: We changed to: “potassium hydroxide”.

L223. Please provide the model and manufacturer name for the ICP-MS

RESPONSE: In the current study, we used ICP-OES instead ICP-MS and the model and manufacturer were mentioned in (Line 231-232): “Thermo Scientific iCAP 7000 Series ICP-OES spectrometer (Thermo Scientific, Cambridge, UK)”.

L235. Full name for Na2CO3 and H2O

RESPONSE: Thanks for the comments, it has been revised. We changed to: (Line 281): “sodium carbonate” and (Line 282): “water”.

L247. Please justify why ascorbic acid calibration curve was utilised to represent the DPPH result

RESPONSE: The use of a calibration curve performed with a reference antioxidant, namely trolox or ascorbic acid, is a recognize method to calculate the scavenging effect on the DPPH radical.  This methodology has been used by several authors and is widely found in the scientific literature (Zang et al., https://doi.org/10.1016/j.foodchem.2016.11.036; Seck et al., https://doi.org/10.1016/j.sajb.2020.11.019; Zhang et al., https://doi: 10.1016/S2095-3119(17)61664-2; Jansen et al., https://doi.org/10.1016/j.foodhyd.2018.09.004; Andrade et al., https://doi.org/10.1016/j.foodres.2017.08.066; Liao et al., doi: 10.4103/0973-1296.96570).

Section 2.4. It would be useful if the authors could find the weather data (e.g. temperature, humidity, rainfall, sunshine hours, soil depth and moisture....) for the different places on the island in the last 3-5 years to identify if weather also play a role in influencing their chemical composition. Please add the information if available. Could be useful for the discussion section as a strong supporting evidence.

RESPONSE: We recognize that this is an important comment, and we would like to clarify that weather data (temperature, humidity and rainfall) has already been made available for Cabo Verde in a former paper published by some of the authors (Table 1, in Monteiro, F.; Fortes, A.; Ferreira, V.; Pereira Essoh, A.; Gomes, I.; Correia, A.M.; Romeiras, M.M. Current Status and Trends in Cabo Verde Agriculture. Agronomy 2020, 10, 74. https://doi.org/10.3390/agronomy10010074). Concerning data available for Cabo Verde soils, this is relatively scarce for this archipelago, but in general soils are poorly evolved, thin and very stony, covering half of the country’s surface, as was also referred by Monteiro et al. (2020).

In line with the Reviewer’s concerns, the section 4.4. “Food security and pulses in Cabo Verde” of the Discussion section has been improved and modified in order to solve these shortcomings (Lines:579-595).

Finally, in order to fully address the concerns of the Reviewers, some references have been added particularly in the Discussion section:

Essoh, A.P.; Monteiro, F.; Pena, A.R.; Pais, M.S.; Moura, M.; Romeiras, M.M. Exploring glucosinolates diversity in Brassicaceae: A genomic and chemical assessment for deciphering abiotic stress tolerance. Plant Physiology and Biochemistry 2020, 150 (2020): 151-161.

https://doi.org/10.1016/j.plaphy.2020.02.032.

Romeiras, M.M.; Vieira, A.; Silva, D.N.; Moura, M.; Santos-Guerra, A.; Batista, D.; Duarte, M.C; Paulo, O.S. Evolutionary and biogeographic insights on the Macaronesian Beta-Patellifolia species (Amaranthaceae) from a time-scaled molecular phylogeny. PLOS ONE 2016, 11(3): e0152456. https://doi.org/10.1371/journal.pone.0152456.

Rocha, V.; Duarte, M.C.; Catarino, S.; Duarte, I.; Romeiras, M.M. Cabo Verde’s Poaceae flora: a reservoir of Crop Wild Relatives diversity for crop improvement. Frontiers in Plant Science 2021, doi: 10.3389/fpls.2021.630217.

L254. Please clarify the term "expert-wise approach"

RESPONSE: We agree that this was not clear in the first version, thus we have revised it to clarify the text that now reads: “The information on economic and agricultural profiles of each bean species under study was also investigated during the field surveys made between 2018 and 2019 at the main trade markets of Santiago Island” (Line 300-302).

Fig 1 and Table 1. Please provide the images for the legumes and their common name.

RESPONSE: We agree with the Reviewer and all the proposed alterations have been incorporated in the Supplementary files (see Supplementary Figure S1).

Figure S1. Are these different places on the same altitude? Please add the information if available.

RESPONSE: We agree with the Reviewer and the data concerning the altitude have been incorporated in the Supplementary files (see Supplementary Table S1).

We hope that these improvements have adequately addressed the Reviewers concerns.

Sincerely,

Maria Romeiras & Maria Paula Duarte